# Mid-Thoracic Vertebral Compression Fracture after Mini-Trampoline Exercise: A Case Series of Seven Patients

**DOI:** 10.3390/medicina59091529

**Published:** 2023-08-24

**Authors:** Sung Cheol Park, Hyoung-Bok Kim, Hoon-Jae Chung, Jae Hyuk Yang, Min-Seok Kang

**Affiliations:** 1Department of Orthopedic Surgery, Bumin Hospital Seoul, Seoul 07590, Republic of Korea; osspinepark@gmail.com (S.C.P.); esshappy@daum.net (H.-B.K.); neoz0708@gmail.com (H.-J.C.); 2Department of Orthopedic Surgery, Korea University Anam Hospital, College of Medicine, Korea University, Seoul 02841, Republic of Korea; kuspine@korea.ac.kr

**Keywords:** thoracic vertebrae, compression fracture, mini-trampoline, case series

## Abstract

*Background and Objectives*: Home-based training exercise gained popularity during the coronavirus disease 2019 pandemic era. Mini-trampoline exercise (MTE) is a home-based exercise that utilizes rebound force generated from the trampoline net and the motion of the joints of the lower extremities. It is known to be beneficial for improving postural balance, stability, muscle strength and coordination, bone strength, and overall health. However, we encountered several patients with mid-thoracic vertebral compression fractures (VCFs) following regular MTE, which was never reported previously, despite having no history of definite trauma. This study aims to report mid-thoracic VCFs after regular MTE and arouse public attention regarding this spinal injury and the necessity of appropriate prior instructions about the correct posture. *Patients and Methods*: All consecutive patients diagnosed with acute VCFs following regular MTE were included. We collected data on patient demographics, history of MTE, characteristics of symptoms, and radiological findings such as the location of fractures and anterior vertebral body compression percentage. *Results*: Seven patients (one man and six women) and ten fractures (T5 = 1, T6 = 3, T7 = 2, and T8 = 4) were identified. Symptoms started 2.57 ± 1.13 weeks after the beginning of regular MTE. All patients reported that they were never properly instructed on the correct posture. They also stated that they were exercising with a hunchback posture and insufficient joint motion of the lower extremities while holding the safety bar with both hands, which resulted in increased peak vertical force along the gravity *z*-axis in the mid-thoracic area and consequent mid-thoracic VCFs. *Conclusions*: Mid-thoracic VCFs can occur following regular MTE even without high-energy trauma in case of improper posture during exercise. Therefore, public attention on mid-thoracic VCFs following MTE and the appropriate prior instructions are imperative.

## 1. Introduction

Physical exercise is indicated to combat a sedentary lifestyle and is considered a valuable means to prevent and manage a variety of diseases, such as cardiovascular disease, obesity, diabetes, musculoskeletal disorders, and psychiatric disorders [1,2,3]. It is also recommended for improving physical performance and reducing the risk of falls in elderly patients with osteoporosis [4,5]. The importance of physical exercise is increasing over the years not only for the physiological but also for the psychological and socioeconomic perspectives [6,7].

However, the coronavirus disease 2019 (COVID-19) pandemic led to social distancing measures because of the high transmission rates of the virus and lack of specific treatments, resulting in decreased physical activity [8,9]. COVID-19 was reported to restrict various types of outdoor physical activities, such as walking, biking, and recreational activity [8]. This may also lead to a negative impact on the overall public and global health status. Subsequently, home-based training exercises were suggested as alternatives [10].

Mini-trampoline exercise (MTE) is a type of home-based training exercise in which one jumps on a mini-trampoline and utilizes the rebound force generated from the net and repeated flexion and extension of the hip joint while holding the safety bar with both hands with the back in an upright position. MTE, which is a low-impact plyometric exercise, is becoming a very popular workout and is known to be highly beneficial for postural balance, stability, muscle strength and coordination, bone remodeling and mineralization, and pelvic floor muscle function in the elderly population [11,12].

Over the past decades, several previous studies investigated trampoline-related injuries from both jump park and home trampolines; reported injuries include head injuries, lacerations, sprains/strains, fractures, or joint dislocations of the extremities [13,14,15]. Regarding spinal injuries, cervical spine fractures and resulting quadriplegia were reported to be mainly the result of somersault or fall injuries [16,17]. However, we encountered several patients with mid-thoracic vertebral compression fractures (VCFs) following MTE despite having no high-energy trauma. To the best of our knowledge, such cases were not reported previously. This article aims to report cases of mid-thoracic VCF after repetitive MTE and arouse public attention regarding the possibility of this injury and the necessity of appropriate prior instructions to prevent the occurrence of such injuries.

## 2. Patients and Methods

This retrospective study was approved by the Institutional Review Board (BMH 2022–04-008). Patient consent was waived because of the retrospective study design and anonymity of included individuals. In this case series, we reviewed all consecutive patients diagnosed with acute VCF following regular MTE between July 2019 and September 2022. Patients with systemic diseases that may affect the bone quality, such as systemic infection, inflammatory spondylitis, history of bone metabolic disease, or history of neoplastic disease, or incomplete medical record documentation were excluded.

All included patients underwent plain radiography, magnetic resonance imaging (MRI), and dual-energy X-ray absorptiometry (DEXA). Although they denied any history of definite trauma, focal tenderness was noted in the mid-thoracic area on the physical examination. Plain radiography showed no specific findings or suspicious endplate disruption or loss of anterior vertebral body height (AVH). However, MRI confirmed VCFs with bone marrow edema.

### 2.1. Data Collection and Radiographic Measurement: Anterior Body Compression Percentage

We collected data on patient demographics (sex, age, body weight, height, and body mass index [BMI]), history of MTE schedule, duration of symptoms, pain intensity on the visual analog scale (VAS), and radiological findings. Regarding the radiographic parameter, the anterior vertebral body compression percentage (AVCP) was calculated to evaluate the amount of AVH loss compared to the intact state using standing neutral plain radiography [18]. The AVCP was defined as the ratio of the AVH loss in comparison to the mean value of the AVH of the adjacent unfractured vertebrae (Figure 1).

### 2.2. Statistical Analysis

Descriptive analyses were performed using SPSS Statistics, version 25.0 (IBM Corp., Armonk, NY, USA).

## 3. Results

Seven patients (one man and six women) and ten mid-thoracic VCFs were identified, with a mean age of 62.86 ± 5.27 years. The mean BMI was 24.44 ± 3.39 kg/m^2^ and the mean T-score on DEXA scans was −2.34 ± 0.56. Only four patients were confirmed to have osteoporosis. The mean follow-up period was 7.71 ± 2.62 months (range: 6–13 months).

The levels of VCFs were T5 in one patient, T6 in three patients, T7 in two patients, and T8 in four patients. The mean AVCP was 14.77 ± 5.12%. Symptoms started 2.57 ± 1.13 weeks after the initiation of regular MTE. The mean duration of the MTE was 40.71 ± 11.34 min per session, three to four times a week.

All patients presented with progressively worsening posture-dependent back pain. No neurological symptoms were observed. All patients stated that they never received proper education about correct posture previously, and they exercised with insufficient movement of the hip joints in a hunchback posture while holding a safety bar with both hands.

They were treated conservatively with thoracolumbar spinal orthosis (TLSO) and bone antiresorptive agents such as zoledronate or denosumab. The pain scores improved to ≤2 on the VAS within 3 months of diagnosis in all patients. The baseline characteristics and clinical information of the patients are shown in Table 1.

### 3.1. Case 1 and 2

Two sisters aged 65 and 63 years, respectively, presented to our hospital with upper back pain for 4 weeks (Patient 1 and 2). They started regular MTE for 45 min per session for health care reasons 4 weeks prior to visiting hospital. They were informed at another clinic that no fractures were found on plain radiography. Despite pain medications and physical therapy, their symptoms gradually worsened. MRI performed at our hospital revealed VCFs in the mid-thoracic spine (Figure 2). Their T-scores were −2.1 and −1.3 on DEXA scans, respectively. They were treated conservatively with TLSO and an intravenous infusion of zoledronic acid (5 mg). Patients’ symptoms decreased from 7 to 1 on the VAS after 1 month.

### 3.2. Case 3

A 53-year-old female patient with upper back pain for 4 weeks visited our hospital (Patient 6). She performed the MTE for 30 min per session four times a week. MRI revealed acute VCFs with bone marrow edema at T6 and T7 (Figure 3). She was confirmed as osteoporosis on DEXA scans. She was also managed with TLSO and intravenous infusion of zoledronic acid (5 mg). Pain resolved from 6 to 2 on the VAS after 2 months from the first visit.

## 4. Discussion

Physical exercise is known to be beneficial for health promotion in people of all ages and sexes. Home-based exercises are easy to perform in daily life and rapidly gained popularity during the COVID-19 pandemic [10]. However, excessive and inappropriate exercise can be harmful, particularly in individuals who are not habitual to regular exercise [19]. Trampoline jumping was reported to be associated with various injuries, including sprains/strains or fractures of the extremities, cervical spine fractures and resulting quadriplegia, lacerations, and head injuries [13,14,20]. However, cases of VCF after MTE, which is a modified type of trampoline and popular home-based exercise, were not previously reported. The objective of this study was to report mid-thoracic VCFs following MTE, despite no history of definite trauma such as falls, collisions with another person, or somersaults. We also aimed to raise public awareness regarding this issue.

MTE is performed using the rebound force generated from the trampoline net with the repeated motion of the joints of the lower extremities. It was reported to be effective in improving postural balance, functional mobility, gait performance, strength, and fear of falling in patients with osteopenia [11]. It was also found to be beneficial for overall health, including anthropometric profile, body composition, lipid and glucose profiles, work capacity, and quality of life [21,22]. In addition, exercises with high-impact loadings, such as jumping, can be helpful for individuals with low bone mineral density because accompanying physical strain would induce bone modeling, remodeling, and mineralization [23].

Various injury types following trampoline exercises were previously reported. Injuries occurring following the use of home and mini-sized trampolines had lower surgery rates and lower percentages of fractures/dislocations and lower extremity fractures than those occurring following the use of park and full-sized trampolines [13,14]. Spinal fractures were also reported after trampoline exercises, primarily affecting the cervical spine [16]. The mechanisms of trampoline injuries include collisions with others, awkward landings, falls from a trampoline, and somersaults [14,24]. However, we encountered several patients with mid-thoracic VCFs without the aforementioned trauma history, which was not documented previously.

We encountered a total of seven patients with mid-thoracic VCF(s) following regular MTE. While the patients denied a history of any other physical activity or definite trauma, they had a common history of regular MTE, which they started within a few weeks prior to the hospital visit. Despite the small number of patients and a lack of evidence of the direct causality due to our study design, a retrospective observational case series, the correlation between a history of repetitive MTE and mid-thoracic VCF may be noteworthy. A future systematic study with a large number of patients or comparative assessment should be conducted to investigate the causal relationship between mid-thoracic VCF and repetitive MTE.

Osteoporotic fracture is a complication of osteoporosis, in which the bone becomes more fragile due to bone deterioration or low bone mass. In our study, only four patients had osteoporosis according to their DEXA scans. This result suggests that mid-thoracic VCF can occur in individuals without extremely poor bone quality during uneventful regular MTE. In other words, these patients’ VCFs may be a form of stress fracture resulting from repetitive usual load rather than the fractures purely caused by bone fragility.

VCFs can remain undiagnosed when clinically asymptomatic or vague [25]. Moreover, mid-thoracic VCFs are relatively more prone to be undiagnosed because it is difficult to discriminate the actual VCF from mild wedge deformities or normal variations in the mid-thoracic spine [26]. Nine of the ten fractured vertebrae of our patients had an AVCP of less than 20%, and several patients stated that they had vague discomfort rather than definite pain at first. Since patients with mid-thoracic VCFs following MTE had obscure symptoms and minimal vertebral body compression in plain radiography, judicious examination and work up are recommended to avoid the missed diagnosis. It may be also imperative in other types of physical and home-based exercise.

All patients stated that they were never properly instructed on the correct method to perform MTEs. They also reported that they were exercising with a hunchback posture and insufficient flexion and extension of the joints of the lower extremities. The coordination dynamics of jumping exercises accompany the motion of the joints of the lower extremities, including the hip, knee, and ankle joints (Figure 4A) [27,28]. These dynamics may dampen the peak vertical force during jumping on the trampoline surface [28]. Meanwhile, it would be better to maintain the spinal angle to maintain balance during a cycle. However, hopping with insufficient joint motion may result in an increased peak vertical force against the surface. Moreover, keeping both hands on the safety bar with insufficient joint motion of the lower extremities can lead to a hunchback posture, which might result in mechanical loading along the gravity *z*-axis (Gz) in the mid-thoracic area (Figure 4B). Repeated Gz forces on the inclined vertebrae can act as shearing and compressive forces, resulting in VCFs even without high-energy trauma (Figure 5). However, the above-described injury mechanism is hypothetical, and there is a lack of official instructions on the posture. Additionally, it is unreasonable to conclude that the education can prevent the mid-thoracic VCF after regular MTE because there was no control group with posture and exercise instruction. A well-designed biomechanical analysis or comparative study regarding the posture is required to verify the exact injury mechanisms with regard to postures, mechanical forces, and vertebral injuries.

This study has several limitations. First, the sample size in this study is too small to generalize our results as widespread conditions. Second, owing to the retrospective nature of this case series, there may be confounding factors that were not considered. Third, the above-mentioned descriptions of the correlation between posture and mechanical force are essentially theoretical rather than practical. Finally, the correlation between the mid-thoracic VCF and regular MTE was relatively low because it is difficult to prove the direct causal relationship in our study. Despite these limitations, this study is, to our knowledge, the first study to report mid-thoracic VCF following regular MTE in patients without a history of definite trauma. Moreover, we expect that this study will arouse public concern about the necessity of proper instructions for safe MTE.

## 5. Conclusions

Home-based exercise is becoming popular during the COVID-19 pandemic era. Mid-thoracic VCFs can occur in elderly individuals following regular MTE, a type of home-based exercise, even without high-energy trauma if improper posture is adopted during exercise. Public attention on this type of spinal injury following MTE and the need for appropriate instructions regarding correct posture and exercise are desired to prevent the occurrence of such injuries in the future.

## Figures and Tables

**Figure 1 medicina-59-01529-f001:**
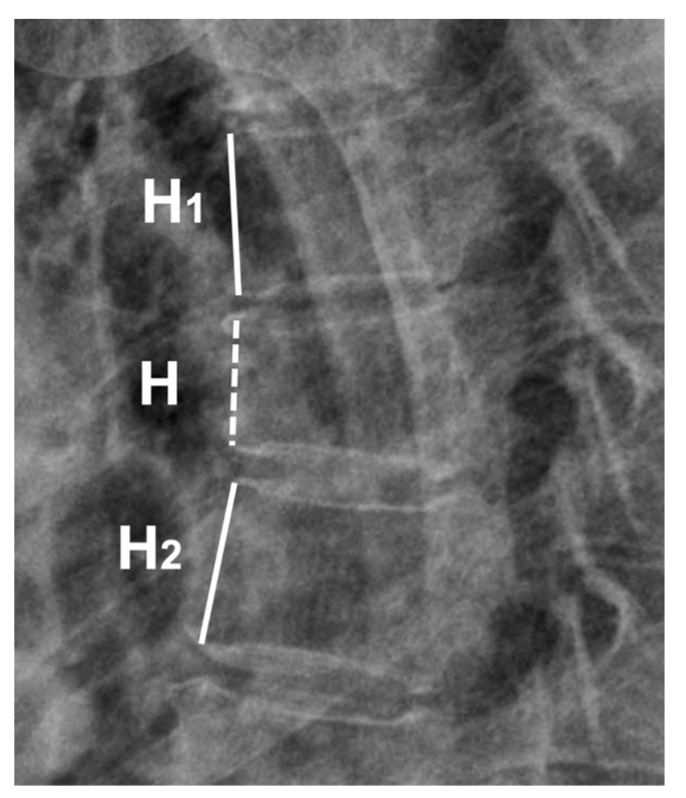
The calculation of anterior vertebral body compression percentage (AVCP). H_1_ represents the anterior vertebral body height (AVH) of the above vertebra. H, AVH of the index vertebra. H_2_, AVH of the below vertebra. AVCP = 100 ∗ [(H_1_ + H_2_)/2 − H]/(H_1_ + H_2_)/2.

**Figure 2 medicina-59-01529-f002:**
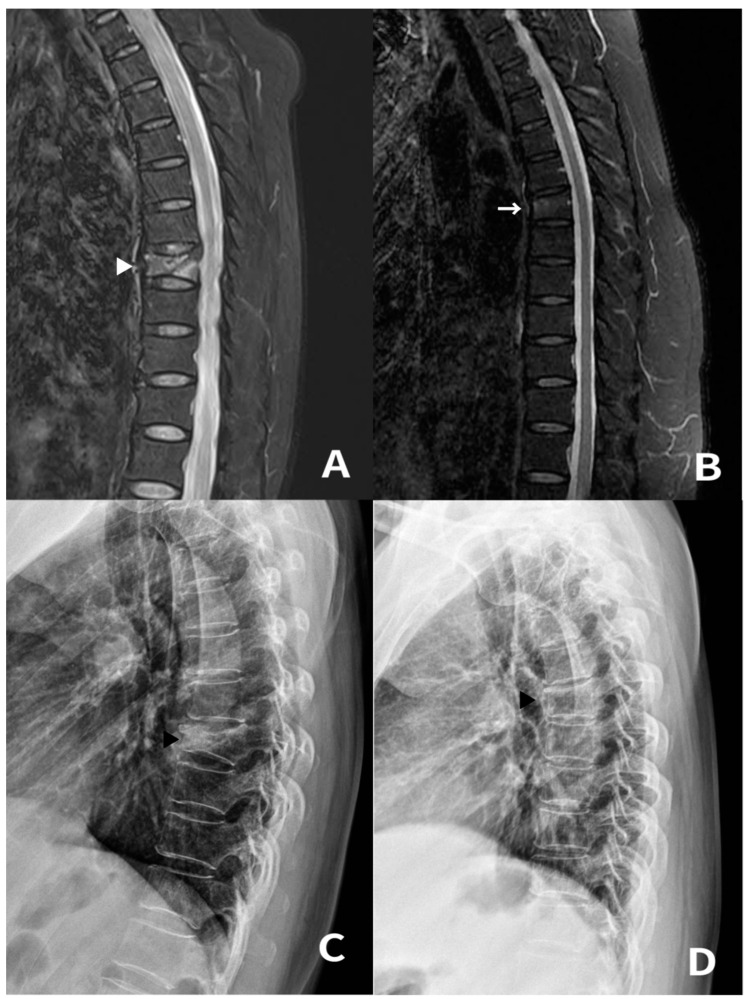
The 65-year-old (Patient 1) and 63-year-old (Patient 2) female patients diagnosed with acute mid-thoracic vertebral compression fracture (VCF) following mini-trampoline exercise (MTE). (**A**) Sagittal fat suppression T2-weighted MR image of Patient 1 showing VCF with bone marrow edema involving T8 (white arrowhead). (**B**) Sagittal fat suppression T2-weighted MR image of Patient 2 showing VCF with bone marrow edema involving T6 (white arrow). (**C**,**D**) The last follow-up X-ray showing well-maintained vertebral bodies without significant height loss (black arrowhead).

**Figure 3 medicina-59-01529-f003:**
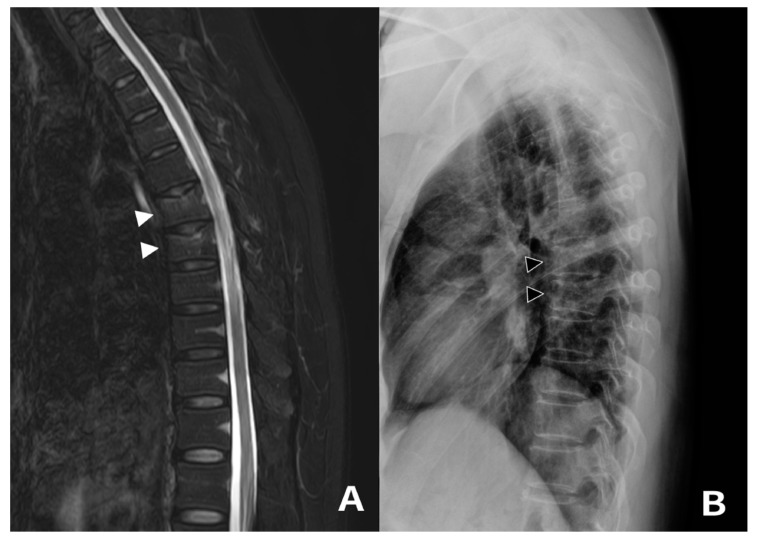
A 53-year-old (Patient 6) female patient diagnosed with acute T6 and T7 vertebral compression fracture (VCF) following mini-trampoline exercise (MTE). (**A**) Sagittal fat suppression T2-weighted MR image showing VCF with bone marrow edema involving T6 and T7 (white arrowheads). (**B**) The last follow-up X-ray showing well-maintained vertebral bodies without significant height loss (black arrowheads).

**Figure 4 medicina-59-01529-f004:**
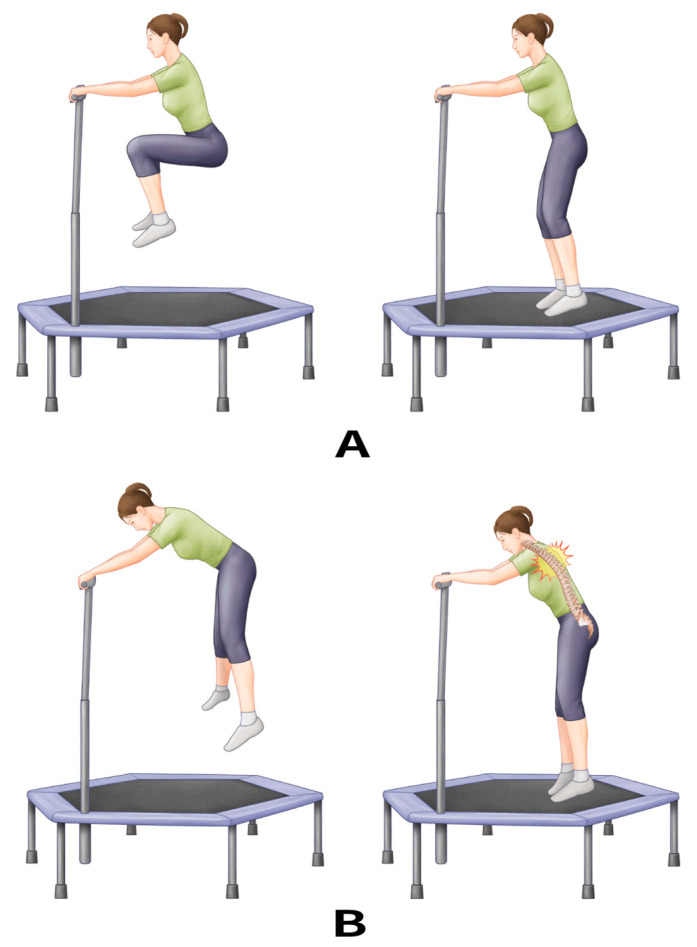
(**A**) Mini-trampoline exercise (MTE) with sufficient joint motion of the lower extremities and maintaining spinal angle. (**B**) MTE with insufficient joint motion of the lower extremities and a hunchback posture.

**Figure 5 medicina-59-01529-f005:**
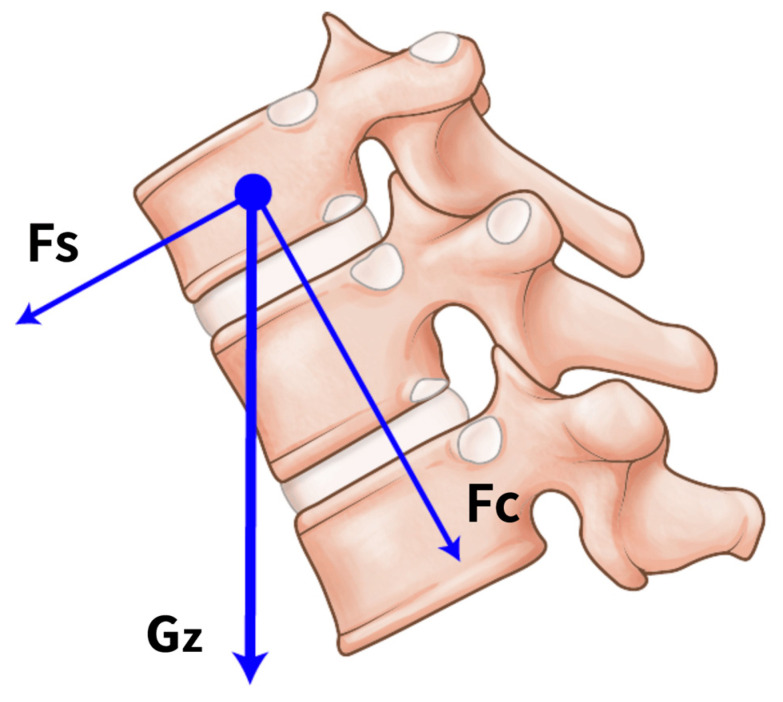
Schematic representation of the gravity *z*-axis (Gz) force on the inclined vertebrae being decomposed into shearing (Fs) and compressive (Fc) forces, resulting in a vertebral compression fracture.

**Table 1 medicina-59-01529-t001:** Baseline characteristics and clinical information of patients.

Patient	Age	Sex	Height (cm)	Weight (kg)	BMI (kg/m^2^)	T-Score	Pain Onset (Weeks)	MTE Duration (min)	Fracture Location	AVCP (%)	VAS	
											Initial	Last
1	65	F	161	61.3	23.65	−2.1	4	45	T8	26.06	7	1
2	63	F	159	79.7	31.53	−1.3	4	45	T6	15.11	7	1
3	60	M	166	67.8	24.60	−2.6	2	60	T8	9.04	8	1
4	64	F	158	60.3	24.15	−2.2	1	45	T8	10.00	7	2
5	70	F	158	57.4	22.99	−3.1	2	30	T7	13.07	7	2
6	53	F	164	55.2	20.52	−2.6	3	30	T6	16.39	6	2
									T7	19.94		
7	65	F	161	61.2	23.61	−2.5	2	30	T5	14.83	6	1
									T6	11.95		
									T8	11.28		

BMI, body mass index; MTE, mini-trampoline exercise; AVCP, anterior vertebral body compression percentage; and VAS, visual analog scale.

## Data Availability

The data presented in this study are available on reasonable request from the corresponding author.

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
