# Peer review of "Mid-Thoracic Vertebral Compression Fracture after Mini-Trampoline Exercise: A Case Series of Seven Patients"

_medicina, 2023, doi:10.3390/medicina59091529_

Round 1

Reviewer 1 Report

"Appropriate instructions regarding correct posture and exercise are necessary to prevent the occurrence of such injuries in the future." There is no control group here, so it is difficult to say that the injuries could have been prevented by posture and exercise instructions.

Author Response

We earnestly appreciate your critical comment and completely understand your concern. It is difficult to conclude that the described VCFs could have been prevented by prior instructions about the correct posture. In addition, the described injury mechanism is hypothetical. However, all included patients state that they had been exercising with a hunch-back posture and insufficient joint motions. Therefore, the need for appropriate instructions regarding correct posture is desired to decrease the risk of such injuries in the future. We added this explanation in the Discussion section (page 6, line 213-219).

Reviewer 2 Report

The cases reported are good described as the study proposal but the correlation between the exercise described, it execution  and the fracture is low.

In elderly patients with already altered T score bone is difficult to prove that the type of exercise described is the cause of fracture.

For this reason, as also underlined by the authors, the work is little strong .

Author Response

We appreciate all your kind comments. We completely agree with you opinion that the direct causality cannot be proven due to our study design. However, although the patients denied a history of any other physical activity or definite trauma, they had a common history of regular mini-trampoline study they had just recently started. It may be meaningful to report the patients with mid-thoracic VCF following regular MTE as a case series. We added this explanation in the Discussion section (page 6, line 177-184) and mentioned this as a limitation (page 8, line 233-235).

Reviewer 3 Report

Although it has some limitations, I think that this study is beneficial in terms of attracting the attention of medical professionals to mid-thoracic VCFs after repetitive Mini-trampoline exercises. It was seen as a valuable study especially in terms of the importance of patient selection and the implementation of appropriate exercise programs for taking necessary precautions.

Author Response

We appreciate the reviewer’s valuable comment and deeply agree with the reviewer’s opinion. 
As the reviewer commented, this study may attract the attention of the public and medical professionals for mid-thoracic VCFs after repetitive Mini-trampoline exercises despite several limitations. 

Round 2

Reviewer 2 Report

Paper  well written. It seems that the authors has made an effort to address the reviewer's opinions on this revised manuscript. 

Manuscript seems more precise and with more information about this particular type of fracture, paying attention about home recrational activity that could be dangerous